# Compact Neural Volumetric Video Representations with Dynamic Codebooks

**Haoyu Guo**[1]    **Sida Peng**[1†]    **Yunzhi Yan**[1]    **Linzhan Mou**[1]
**Yujun Shen**[2]    **Hujun Bao**[1]    **Xiaowei Zhou**[1]

[1]Zhejiang University    [2]Ant Group

## Abstract

This paper addresses the challenge of representing high-fidelity volumetric videos with low storage cost. Some recent feature grid-based methods have shown superior performance of fast learning implicit neural representations from input 2D images. However, such explicit representations easily lead to large model sizes when modeling dynamic scenes. To solve this problem, our key idea is reducing the spatial and temporal redundancy of feature grids, which intrinsically exist due to the self-similarity of scenes. To this end, we propose a novel neural representation, named dynamic codebook, which first merges similar features for the model compression and then compensates for the potential decline in rendering quality by a set of dynamic codes. Experiments on the NHR and DyNeRF datasets demonstrate that the proposed approach achieves state-of-the-art rendering quality, while being able to achieve more storage efficiency. The source code is available at https://github.com/zju3dv/compact_vv.

## 1    Introduction

Volumetric videos record the content of dynamic 3D scenes and allow users to watch captured scenes from arbitrary viewpoints, which has a wide range of applications, such as virtual reality, augmented reality, telepresence, and so on. Traditional methods typically represent volumetric videos as a sequence of textured meshes, which are reconstructed by the multi-view stereo [36, 37, 12, 15] or depth fusion [10, 27]. Such techniques require sophisticated hardware, *e.g.*, multi-view RGB-D sensors, to achieve high reconstruction accuracy, which is expensive and limits their application scenarios.

Recently, implicit neural representations [24, 1, 49, 2] have emerged as a promising strategy to recover 3D scenes from only 2D images. As a representative work, neural radiance field (NeRF) [24] employs an MLP network to predict the density and color for any 3D point, and effectively learns the model parameters from 2D images through the volumetric rendering technique. DyNeRF [17] extends NeRF to dynamic scenes by introducing time-varying latent codes as an additional input of the MLP network, thereby enabling the model to encode the per-frame content of the scene. Although these methods achieve impressive rendering quality, they are extremely slow to train, especially in dynamic scenes. For example, DyNeRF requires over 1000 GPU hours to learn a 10-second volumetric video.

To overcome this problem, some methods exploit explicit representations, such as feature volumes [8, 50, 20] or feature planes [35, 4, 38], to accelerate the training process. They typically store learnable feature vectors at explicit structures and use the interpolation technique to assign a feature

---

[†]Corresponding author.

37th Conference on Neural Information Processing Systems (NeurIPS 2023).

vector to arbitrary scene coordinate, which is then fed into a lightweight MLP network to predict the density and color. By decreasing the number of parameters in the MLP network, these methods achieve a significant speedup in training. However, the explicit representations can easily take up large storage, which is hard to scale to dynamic 3D scenes.

Some methods [42, 16] reduce storage redundancy based on the internal similarity of scenes, achieving satisfactory compression results on static scenes. However, they do not take into account the characteristics of dynamic scenes. Experiments show that directly applying these methods to dynamic scenes results in non-negligible performance degradation. For dynamic scenes, we observe that the redundancy of features learned by existing methods [8, 50, 20, 35, 4, 38] is not only reflected in spatial correlations, but more importantly, dynamic scenes are essentially produced by the movement and changes of static scenes, thereby intrinsically having a strong temporal correlation. If this prior knowledge is not adequately utilized, the model will inevitably contain a considerable amount of unnecessary repetitive storage.

Motivated by these observations, we propose a novel representation, named **dynamic codebook**, for efficient and compact modeling of dynamic 3D scenes. We first employ a compact codebook to accomplish compression, then incrementally update the codebook to compensate for the potential decline in rendering quality of dynamic detailed regions caused by compression. The representation of codebook can effectively mitigate the issue of code redundancy in time and space, thereby significantly reducing unnecessary storage. However, this representation might cause the codes of some regions to be less precise, which will decrease the rendering quality, especially in dynamic detailed regions. The spatial distribution of these regions may vary significantly over time. Therefore, we separately identify the most necessary parts of the codes to optimize for each small time fragment and incrementally add them to the codebook to improve the rendering quality of these regions. Since these regions typically account for a small proportion of the scene, the new codes will not occupy much storage. Therefore, our proposed dynamic codebook based strategy can achieve a high compression rate while maintaining high rendering quality.

We evaluate our approach on the NHR [48] and DyNeRF [17] datasets, which are widely-used benchmarks for dynamic view synthesis. Experimental results demonstrate that the proposed approach could achieve rendering quality comparable to the state-of-the-art methods while achieving much higher storage efficiency.

In summary, our contributions are as follows: (1) We propose a novel representation called dynamic codebook, which effectively decreases the model size by reducing the feature redundancy in time and space while maintaining the rendering quality of dynamic detailed regions. (2) Experiments demonstrate that our approach achieves rendering quality on par with state-of-the-art methods, while significantly improving storage efficiency.

## 2 Related works

**View synthesis of static scenes.** Traditional methods use mesh to represent scene geometry and texture mapping to depict the scene's appearance, which can be rendered from arbitrary viewpoints. However, the reconstruction process based on this representation is typically not end-to-end, leading to a lack of photo-realism of the rendering results. [24] proposes a continuous representation that models the density and color of static scenes with MLP networks, which can be used for high-quality free view synthesis using volume rendering technique. [1] utilizes multi-scale representation to address the inherent aliasing of NeRF, and improve the ability to represent fine details. [49, 2] enable modeling of unbounded scenes with complex backgrounds by the design of coordinate mapping.

The pure MLP-based representation is much more compact compared to traditional representations, however, a large amount of points sampling is required during rendering, and each point will be fed into the MLP, resulting in very low efficiency. [21] utilizes a sparse occupancy grid to avoid sampling points in empty space to accelerate the training and rendering speed. [32] replaces the large MLP in NeRF with thousands of tiny MLPs. [9, 41] directly adopt the explicit volumetric representation during training and utilize coarse to fine strategy to achieve high resolution. [25] employs multiresolution hash table for encoding and lightweight MLPs as decoders, achieving unprecedentedly fast training. [3] further extends anti-aliasing to grid-based representation by multisampling and prefiltering. [33] introduces a novel piecewise-projective contraction function to model large unbounded scenes with feature grids, and enables real-time view synthesis.

**Volumetric video.** Following NeRF's considerable success in modeling static scenes, some studies [19, 30, 17, 31, 35, 45] have sought to extend it to dynamic scenes. There are primarily two approaches to this. The first approach is to directly model temporal variations by directly modifying the input format of MLP to enable view synthesis not only at any viewpoint but also at any given time. [18] directly input the timestamp to MLP for space-time view synthesis. [17] utilizes time-varying latent codes as the additional input to MLP, the latent codes can be learned to model the per-frame content of dynamic scenes. The training of this approach is typically challenging, either requiring some additional supervision such as monocular depth or optical flow, or taking much time to converge. The second approach involves using NeRF to model a canonical space and then learning the deformation from any moment to this canonical space. [31] employs another MLP to model the deformation at each timestamp to this canonical space. [28] utilizes latent codes to model deformation. However, deformation struggles to model complex temporal changes, such as large movements, or the emergence of new content, such as fluid mixing Moreover, these two kinds of approaches both encounter issues similar to those faced by NeRF in static scenes, namely, low training and rendering efficiency.

Methods based on explicit or hybrid representations have also been studied and explored for modeling volumetric videos. [22] encodes multi-view information at arbitrary timestamp and uses a 3D deconvolutional network to produce volumes of density and color. [47] uses 3D CNN to predict feature volumes instead of the explicit volumes, and additionally adopts an MLP to predict the radiance fields. [8, 20, 50] employ a feature grid to model the canonical space, which enables efficient training and rendering. [35, 4, 38] represent volumetric videos using multiple feature planes, offering a more compact representation compared to the feature grids. [45] explicitly partitions the scene into dynamic and static regions using proposed variation field, and models them separately using TensoRF [5], allowing for remarkably fast training. [40] divides scenes into static, deforming and new regions, and utilizes a sliding window-based hybrid representation for efficient scene modeling. [46] explicitly models the residual information between adjacent timestamps in the spatial-temporal feature space and employs a compact motion grid along with a residual feature grid to exploit inter-frame feature similarities.

**Compression of implicit neural representations.** Methods based on feature grids and feature planes can provide significant efficiency improvements, but they also introduce substantial storage requirements. To alleviate this issue, [7] proposes to use pruning strategy to eliminate unnecessary parameters in the feature grid. [5] decomposes the 3D grids into lower dimension. It utilizes VM decomposition to decompose the 3D grid into multiple 2D matrices and even employs CP decomposition to decompose it into multiple 1D vectors. While CP decomposition significantly improves storage efficiency, it results in a non-negligible performance loss. [43] implements compression that can be dynamically adjusted by the proposed rank-residual learning strategy. Vector quantization [11, 13] is a classic compression technique that is widely applied in 2D image and video compression tasks [26, 6, 39]. The main idea of which is to merge similar codes into one through clustering, thereby reducing storage redundancy. Recently, VQAD [42] and VQRF [16] use vector quantization to achieve compression of static 3D scenes. VQAD learns a codebook while training the model, while VQRF implements compression in a post-processing manner. [14] utilizes Fourier transform for compression, which is able to capture high-frequency details while remaining compact. [34] proposes to use wavelet coefficients to improve parameter sparsity.

## 3 Method

Given multi-view videos captured by synchronized and calibrated cameras, our goal is to generate a volumetric video that requires low disk storage while retaining the capability of high-fidelity rendering. Our approach is illustrated in Fig. 1. We represent the volumetric video by multiple feature planes, which can be learned from multi-view videos using the volume rendering technique (Sec. 3.1). To enhance the storage efficiency of the model, we propose a codebook-based representation, which aims to reduce the redundancy of the feature plane, thereby achieving model compression (Sec. 3.2). To compensate for the loss of rendering quality in certain areas due to compression, we extend the codebook to a dynamic codebook. This significant enhancement in rendering quality introduces only a minimal amount of additional storage (Sec. 3.3).

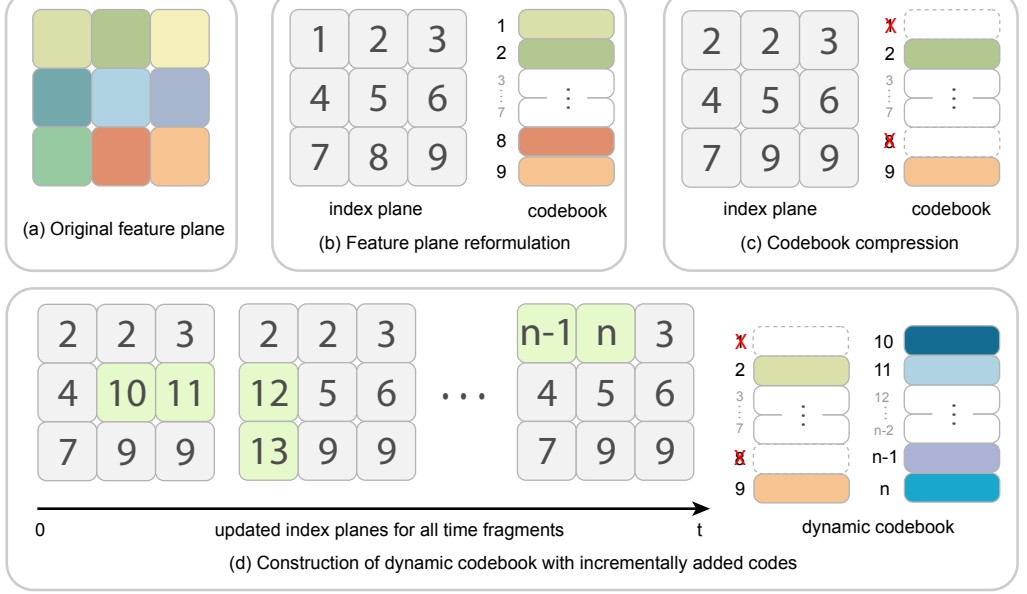

Figure 1: **Overview of our method. (a)** We model volumetric videos with multiple feature planes, which can be learned with the supervision of multi-view 2D videos. **(b)** To facilitate the subsequent process, we simply convert the feature planes to an index plane as well as a codebook, the codebook is constructed by directly flattening the feature plane. **(c)** To reduce the model redundancy, we use a clustering method to merge similar codes, such as code $\{1, 2\}$ and $\{8, 9\}$ in (b). **(d)** To compensate for the decline in rendering quality caused by the compression, dynamic codes are appended to the trimmed codebook. First, we maintain an index plane for each time fragment, where most indices on these planes remain the same. Then, we optimize a portion of the codes (masked as green in this figure) for each time fragment. These codes are incrementally added to the codebook, and the corresponding positions in the index plane are updated, such as $\{10, 11\}$ in the index plane of the first time fragment in this figure.

## 3.1 Learning volumetric videos with feature planes

Inspired by [35, 4, 38], we model the dynamic scene by six feature planes, including three spatial planes denoted as $\mathbf{P}_{xy}$, $\mathbf{P}_{xz}$, and $\mathbf{P}_{yz}$ and three spatial-temporal planes denoted as $\mathbf{P}_{xt}$, $\mathbf{P}_{yt}$, and $\mathbf{P}_{zt}$. Given a 3D coordinate $\mathbf{x} = (x, y, z)$ and time $t$, we obtain six feature vectors $\{\mathbf{f}(\mathbf{x}, t)_i\}$ by normalizing $(\mathbf{x}, t)$ according to the resolution of feature planes and projecting onto six planes:

$$f(\mathbf{x}, t)_i = \psi\big(\mathbf{P}_i, \pi_i(\mathbf{x}, t)\big), \tag{1}$$

where $\pi_i$ denotes projection function to $i$-th feature plane, and $\psi$ denotes bilinear interpolation. We then calculate the feature vector of $\mathbf{x}$ by:

$$\mathbf{f}(\mathbf{x}, t) = f(\mathbf{x}, t)_{xy} \odot f(\mathbf{x}, t)_{zt} + f(\mathbf{x}, t)_{xz} \odot f(\mathbf{x}, t)_{yt} + f(\mathbf{x}, t)_{yz} \odot f(\mathbf{x}, t)_{xt}, \tag{2}$$

where $\odot$ is the Hadamard product. Then $\mathbf{f}(\mathbf{x}, t)$ is fed into an MLP network to predict the density $\sigma(\mathbf{x}, t)$ and color $\mathbf{c}(\mathbf{x}, t)$.

Following [24], we adopt volume rendering to optimize the feature planes as well as the MLP parameters. Specifically, to render a pixel $\mathbf{p}$ at time $t$, we first sample a set of 3D coordinates $\{\mathbf{x}_i\}$ along the ray $\mathbf{r}$ passing through $\mathbf{p}$, and then calculate the accumulated color along the ray:

$$\hat{\mathbf{C}}(\mathbf{r}, t) = \sum_{i=1}^{K} T_i \alpha_i \mathbf{c}_i = \sum_{i=1}^{K} T_i(1 - \exp(-\sigma_i \delta_i))\mathbf{c}_i, \tag{3}$$

where $\delta_i = ||\mathbf{x}_{i+1} - \mathbf{x}_i||_2$ is the distance between adjacent sampled points, and $T_i = \exp(-\sum_{j=1}^{i-1} \sigma_j \delta_j)$ denotes the accumulated transmittance along the ray.

Then we optimize the model with L2 rendering loss with the supervision of ground truth pixel colors. Moreover, we use the standard regularization terms, including L1 norm loss and TV (total variation) loss following [5]. Please refer to supplementary materials for more details.

## 3.2 Compression of model

Though volumetric videos based on feature plane representations allow for efficient training and rendering, they suffer from a substantial model size, which in practice can reach approximately 30-50MB per second. This poses a significant challenge for the storage and dissemination of volumetric videos. In this section, we demonstrate how to achieve model compression using a codebook-based representation.

Previous works in static scene reconstruction and view synthesis have demonstrated that the usage of feature grids or feature planes introduces significant spatial redundancy. They achieve compression by reducing spatial redundancy through CP decomposition [5, 43], vector quantization [42, 16], Fourier / Wavelet transformations[14, 34], and so on. For volumetric videos, redundancy exists not only spatially, but also temporally. The usage of multiple 2D planes has to some extent mitigated the spatial redundancy. However, spatial redundancy still exists, and more importantly, there is non-negligible temporal redundancy.

As stated by [16], when representing 3D scenes using feature grids, $99.9\%$ of the importance is contributed by merely $10\%$ of the voxels. While the situation is not as extreme when we use 2D planes to represent volumetric videos, a similar pattern can still be observed. Hence, we can apply this principle to reduce the redundancy in the feature planes and achieve compression.

To facilitate the subsequent process, we initially convert the model into a codebook format, as illustrated in Fig. 1b. Specifically, we flatten all features on feature planes into a codebook, after which we convert the feature planes into index planes. In other words, we store the index mapping to the codebook at each grid point of the planes, instead of directly storing features.

In order to perform model compression based on the principle discussed above, we first calculate the importance score for each code in the codebook, according to its contribution to the rendering weight in the volume rendering process. Specifically, we randomly sample rays among all camera views and timestamps to perform volume rendering. For each sampled point $\mathbf{p}_i$ on each ray, we record the volume rendering weights $T_i\alpha_i$ of $\mathbf{p}_i$, we also record the corresponding codes as well as their weights when performing trilinear interpolation at point $\mathbf{p}_i$. We calculate the importance score of each code by accumulating the rendering weights of all points that are mapped to it:

$$I_c = \sum_{\mathbf{p}_i \in \mathcal{N}_c} w_{ci} \cdot T_i\alpha_i, \tag{4}$$

where $\mathcal{N}_c$ denotes the set of points that are mapped to code $c$, and $w_{ci}$ denotes the trilinear interpolation weight that code $c$ contribute to point $p_i$. To reduce the storage, we first discard the less important codes by merging a portion of the codes with the lowest importance scores into a single one. For codes with medium importance scores, we cluster them into a smaller number of codes using a clustering method, as illustrated in Fig. 1c. The clustering is performed in an optimization manner, following VQ-VAE [44]. Specifically, we randomly initialize $n$ codes, where $n$ is much smaller than the number of the codes to be clustered. Then we use exponential moving averages (EMA) to update the codes iteratively, please refer to the supplementary materials for more details. We retain the codes with the highest importance scores as they are. After completing the merging and clustering operations, we update the indices on the index planes for each time fragment accordingly.

## 3.3 Dynamic codebook

The codebook-based compression method can significantly reduce model storage but inevitably leads to a decrement in rendering quality. We observe that the rendering of detailed regions will degrade after compression. The reason lies in the fact that these regions require particularly high precision for codes. The previously mentioned clustering strategy may reduce the accuracy of some of these codes. Although the importance score can measure the contribution of each code during the rendering process, it does not distinguish well between its contribution to detailed and non-detailed regions. Although these detailed regions typically occupy a small proportion of the scenes, their

spatial distribution may vary over time, making it difficult to enhance the rendering quality by directly optimizing a small subset of codes.

To solve the problem, we first divide the temporal dimension into numerous fragments. Our goal is to adaptively identify and optimize the parts most in need of enhancement within each time fragment. These are the portions where the shared spatial feature plane provides relatively poorer representation during that time fragment. Since these parts constitute a smaller proportion, they do not lead to a substantial storage increase.

To this end, we leverage the backpropagation gradient to pinpoint the top $k$ codes demanding the most optimization within each time fragment. Specifically, for each time fragment, we run the forward and backpropagation steps in the training process without optimization and accumulate the gradients of each code for several iterations. Then we simply select the $k$ codes with the highest accumulated gradients and optimize them for this time fragment. We repeat this process for each time fragment, and the optimized codes are independent of all time fragments.

Thanks to the codebook representation, we can append the optimized codes for each time fragment into the codebook. Additionally, we store an independent index plane for this time fragment and update the corresponding indices within it, as shown in Fig. 1d. We term the final codebook as **dynamic codebook** because it is constructed incrementally by optimizing for each time fragment.

The index planes store integer indices, thus occupying minimal storage space. Furthermore, as each time fragment only requires optimization of a small number of codes, the dynamic codebook does not introduce large additional storage. However, the design of dynamic codebook could effectively enhance the rendering quality of detailed regions.

**Post-processing.** For further compression, we apply post-processing steps to the codebook. We use uniform weight quantization to quantize the codebook. Then we encode all components, including the quantized dynamic codebook, index planes as well as MLP parameters with entropy encoding [51].

## 4 Implementation details

**Network architecture.** We set the resolution of the feature planes in the spatial dimension as 256 on NHR and 640 on DyNeRF dataset, and the one in the temporal dimension is set as 100 on NHR and 150 on DyNeRF dataset. We model density and appearance with two individual feature planes with the number of feature channels as 16 and 48 respectively. For appearance feature planes, our model also includes multi-scale planes in addition to the six feature planes at the highest resolution to encourage spatial smoothness and coherence, following [35]. For forward-facing scenes in the DyNeRF dataset, we apply NDC transformation that bounds the scene in a perspective frustum.

**Construction of dynamic codebook.** We construct a separate codebook for both density and appearance. During the compression step, we discard the portion of codes that only accumulate an importance score contribution of $0.001\%$, by merging them into a simple zero code. We then retain the top $30\%$ of codes with the highest importance score contributions and cluster the remaining codes into $k$ codes. We set $k$ as 4096 for NHR dataset and 16384 for DyNeRF dataset. We divide each scene on NHR / DyNeRF into 100 / 60 time fragments, and optimize 1000 appearance codes and 5000 density codes for each time fragment.

**Training.** We implement our method with PyTorch [29]. We set the learning rate as 2e-3 for MLP parameters and 0.03 for feature planes, and use Adam optimizer to train the network with batches of 4096 rays. We train the model on one single NVIDIA A100 GPU, which takes about 1.5 / 4.3 hours for training and 1.0 / 1.7 hours for the construction of dynamic codebook on NHR / DyNeRF datasets.

## 5 Experiments

### 5.1 Datasets

To evaluate the performance of our approach, we conduct experiments on NHR [48] and DyNeRF [17] datasets. NHR contains four videos recorded at 30fps that capture an athlete performing large

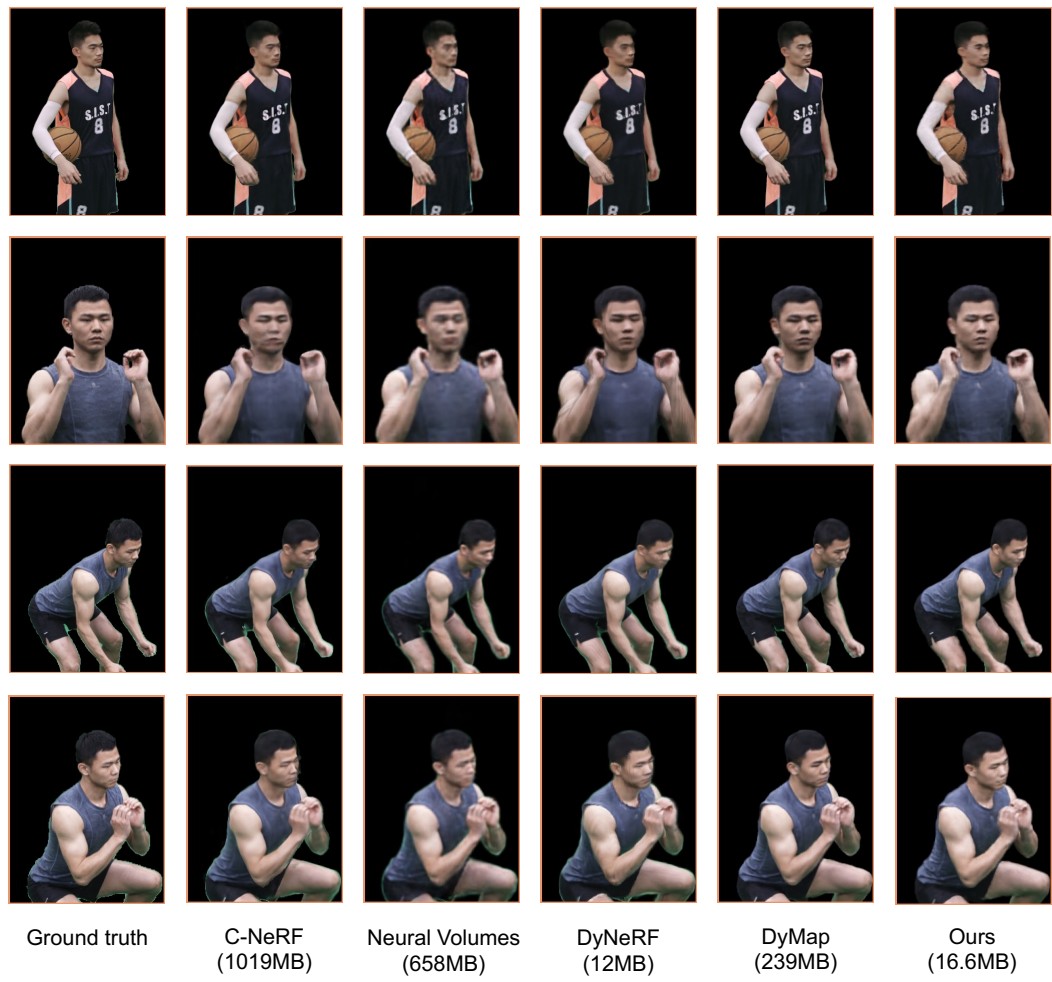

| Ground truth | C-NeRF (1019MB) | Neural Volumes (658MB) | DyNeRF (12MB) | DyMap (239MB) | Ours (16.6MB) |

Figure 2: **Qualitative results on NHR dataset.** Our method achieves comparable rendering quality to state-of-the-art methods while requiring significantly less storage. Note that although DyNeRF also has a small storage size, it requires days for training.

Table 1: **Quantitative results on NHR dataset.** Metrics are averaged over all scenes. Note that D-NeRF and DyNeRF are both single MLP based methods, which is of low model size but extremely slow training speed.

|  | NV[22] | C-NeRF[47] | D-NeRF[31] | DyNeRF[17] | DyMap[30] | K-Planes[35] | Ours |
|---|---|---|---|---|---|---|---|
| PSNR↑ | 30.86 | 31.32 | 29.25 | 30.87 | 32.30 | 31.08 | 32.10 |
| SSIM↑ | 0.941 | 0.949 | 0.920 | 0.943 | 0.953 | 0.946 | 0.947 |
| Size (MB)↓ | 658 | 1019 | 4 | 12 | 239 | 103 | 16.6 |

movements in a bounded scene with simple backgrounds. Following the setting of DyMap [30], we select 100 frames from each video and use 90 percent of camera views for training and the rest for testing. DyNeRF contains six 10 videos recorded at 30fps that capture a person cooking in a kitchen by 15-20 cameras from a range of forward-facing view directions. We use all 300 frames and follow the same training and testing split as in [17].

## 5.2 Comparison with the state-of-the-art methods

**Baseline methods.** For experiments on NHR dataset, we compare with (1) Neural Volumes (NV) [22] (2) C-NeRF [47] (3) D-NeRF [31] (4) DyMap [30] (5) K-Planess [35]. For experiments

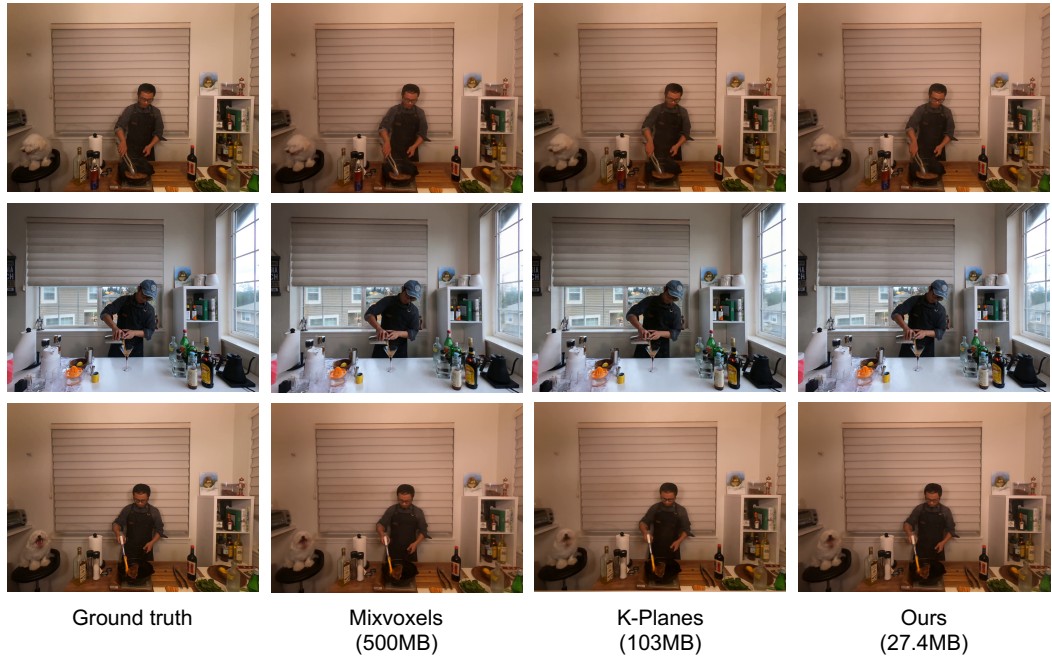

| Ground truth | Mixvoxels (500MB) | K-Planes (103MB) | Ours (27.4MB) |

Figure 3: **Qualitative results on DyNeRF dataset.** Our method achieves comparable rendering quality to previous methods while requiring significantly less storage. Please zoom in for details.

Table 2: **Quantitative results on DyNeRF dataset.** Metrics are averaged over all scenes. DyNeRF[1] and LLFF[1] only report metrics on the flame salmon scene. We provide more detailed results on each scene in the supplementary materials.

|  | LLFF[1] [23] | DyNeRF[1] [17] | Mixvoxels [45] | K-Planes [35] | Ours |
|---|---|---|---|---|---|
| PSNR↑ | 23.24 | 29.58 | 30.81 | 30.54 | 30.58 |
| SSIM↑ | 0.848 | 0.961 | 0.960 | 0.927 | 0.923 |
| Size (MB)↓ | - | 28 | 500 | 103 | 27 |

on DyNeRF dataset, we compare with (1) LLFF (2) The original method proposed in DyNeRF [17] (3) Mixvoxels [45] (4) K-Planes [35]. For K-Planes, we run their official implementation and found that the results on DyNeRF are slightly different from their paper. For other methods, we directly adopt the results from [30] on NHR and [35, 45] on DyNeRF dataset.

Tabs. 1 and 2 list the comparison of our method with the state-of-the-art methods on NHR and DyNeRF datasets in terms of rendering quality and model size. We report PSNR and SSIM as the metrics for rendering quality, and the size of the model is measured in megabytes (MB). We also provide qualitative results in Figs. 2 and 3. The results show that our method can achieve rendering quality comparable to the state-of-the-art while offering significant advantages in storage efficiency. Single MLP-based methods, such as D-NeRF and DyNeRF, have similar storage sizes to ours, but their training process is extremely slow, requiring more than days.

### 5.3 Ablation studies

To analyze the effectiveness of our proposed dynamic codebook strategy, we conduct ablation studies on NHR and DyNeRF datasets. We first train the feature plane based model without compression, then we apply compression without dynamic codebook, and finally, we utilize dynamic codebook to incrementally refine the model. We report quantitative results in Tab. 3 and provide qualitative results in Fig. 4. Results in Tab. 3 show that the compression step can drastically reduce storage requirements, but causes a noticeable decline in rendering quality. And the proposed dynamic codebook strategy can compensate for this by introducing a small amount of additional storage while

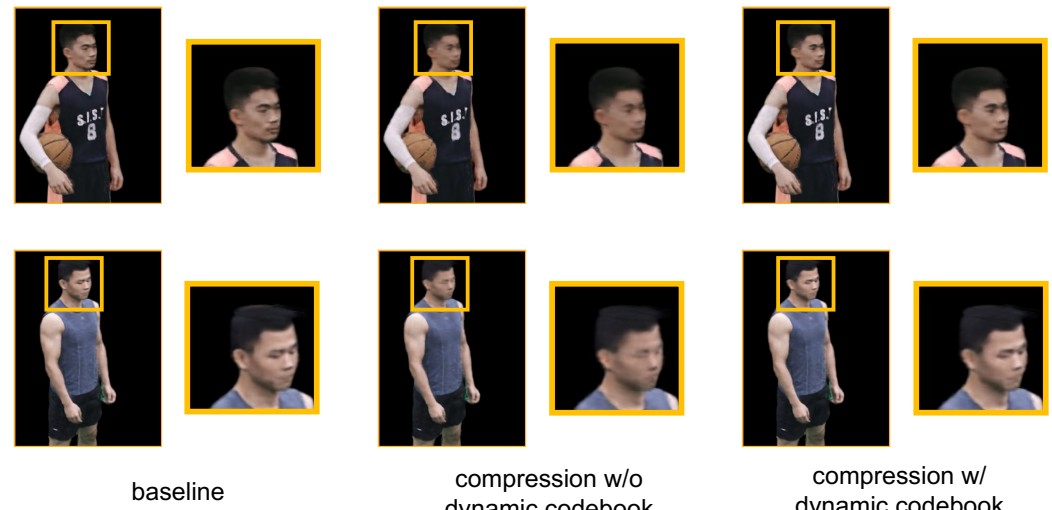

Figure 4: **Qualitative results of ablation studies on NHR dataset.** Compression without dynamic codebook will result in a non-negligible loss of rendering quality in facial regions, while the usage of dynamic codebook can effectively compensate for this.

Table 3: **Quantitative results of ablation studies.** We report the rendering quality and model before compression and after compression with and without DC (dynamic codebook).

| Method | NHR | | | DyNeRF | | |
|---|---|---|---|---|---|---|
| | PSNR↑ | SSIM↑ | Size (MB)↓ | PSNR↑ | SSIM↑ | Size (MB)↓ |
| basline | 32.16 | 0.947 | 91 | 30.56 | 0.923 | 523 |
| compression *w/o* DC | 31.52 | 0.943 | 5.7 | 30.37 | 0.920 | 22 |
| compression *w/* DC | 32.10 | 0.947 | 16.6 | 30.58 | 0.923 | 27 |

achieving rendering quality almost equivalent to the level before compression. Results in Fig. 4 demonstrate that compression particularly degrades the rendering quality in detailed regions, such as the facial regions. The use of a dynamic codebook can effectively compensate for the loss induced by compression.

## 6 Conclusion

We presented a novel neural representation, named dynamic codebook, for learning neural volumetric videos with low storage costs while enabling high-quality view synthesis. Our approach models the volumetric video using a set of feature planes and compresses the video by reducing the spatial and temporal redundancy of features. We calculate the importance score of features and merge similar features that are of small importance scores. To compensate for the artifacts caused by the compression, our approach pinpoints the regions with lower rendering quality and appends dynamic codes for them. Experiments demonstrated that our approach achieves competitive rendering quality while being able to take up much fewer storage cost compared to state-of-the-art methods.

**Discussion.** We currently update the code independently for each time segment, which takes more than one hour for each dynamic scene, thereby slowing down the reconstruction. A solution is to increase the parallelism of optimizing codes. We leave it to future works.

**Acknowledgement.** The authors would like to acknowledge support from NSFC (No.62172364), Information Technology Center and State Key Lab of CAD&CG, Zhejiang University.

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
