# Supplementary Material: Compact Neural Volumetric Video Representations with Dynamic Codebooks

**Haoyu Guo**[1]   **Sida Peng**[1†]   **Yunzhi Yan**[1]   **Linzhan Mou**[1]
**Yujun Shen**[2]   **Hujun Bao**[1]   **Xiaowei Zhou**[1]

[1]Zhejiang University   [2]Ant Group

## A    Regularization to feature planes

In addition to L2 rendering loss, we apply standard regularization. On NHR dataset, we use L1 regularization to all density feature planes to remove floaters and outliers and improve the quality in extrapolating views, which is expressed as:

$$\mathcal{L}_{L1} = \sum_i ||\mathbf{P}_i||, \tag{1}$$

where $\mathbf{P}_i$ is the $i$-th feature plane. On DyNeRF dataset, we use TV (total variation) regularization on both density and appearance feature planes, which is expressed as:

$$\mathcal{L}_{TV} = \sum_i \Delta^2 \mathbf{P}_i, \tag{2}$$

where $\Delta^2 \mathbf{P_i} = \sum_{j,k} ||\mathbf{P}_i^{j,k} - \mathbf{P}_i^{j-1,k}||_2^2 + \sum_{j,k} ||\mathbf{P}_i^{j,k} - \mathbf{P}_i^{j,k-1}||_2^2$ is the squared difference between the neighboring values.

## B    Implementation details of clustering

To cluster the codes with medium importance score, we first randomly initialize $k$ codes $\{c_i, i = 1, ..., k\}$. Then we update the codes using exponential moving average (EMA). Specifically, we perform forward with mini-batches iteratively, and update the codes with the following equation:

$$N_i^{(t)} := N_i^{(t-1)} * \gamma + n_i^{(t)}(1 - \gamma), \tag{3}$$

$$m_i^{(t)} := m_i^{(t-1)} * \gamma + \sum_j z_{i,j}^{(t)}(1 - \gamma), \tag{4}$$

$$c_i^{(t)} := \frac{m_i^{(t)}}{N_i^{(t)}}, \tag{5}$$

where $\{z_{i,j}^{(t)}\}$ denotes the set of features that are mapped to code $c_i$ in iteration $t$, and $\gamma$ is the coefficient of EMA, which is set as 0.8 in our experiments.

## C    Detailed experiment results

We provide detailed experiment results on each scene. Results on NHR dataset are shown in Tab. 4, and results on DyNeRF dataset are shown in Tab. 5.

37th Conference on Neural Information Processing Systems (NeurIPS 2023).

# D   Ablation studies of hyperparameters

Our method indeed introduces some new hyperparameters such as the ratio of codes to discard or retain during codebook compression and the size of the codebook (k). To analyze the impact of these hyperparameters on rendering quality and compression ratio, we designed corresponding ablation studies on NHR, and the results are shown in Table 1 and Table 2. The results evident that both codebook size and the retention ratio influence the storage size and rendering quality of our method. However, the impact is not substantial, indicating that our method is fairly robust to these hyperparameters.

Table 1: **Ablation studies of percent of retained code.**

| percent of retained code | 10 | 20 | 30 |
|---|---|---|---|
| PSNR (w/o dynamic codebook) | 32.39 | 32.57 | 32.85 |
| PSNR (w/ dynamic codebook) | 33.40 | 33.46 | 33.51 |
| final model size (MB) | 16.3 | 16.5 | 16.6 |

Table 2: **Ablation studies of codebook size.**

| k | 1024 | 2048 | 4096 | 8192 | 16384 |
|---|---|---|---|---|---|
| PSNR (w/o dynamic codebook) | 32.09 | 32.47 | 32.85 | 32.89 | 33.01 |
| PSNR (w/ dynamic codebook) | 33.30 | 33.43 | 33.51 | 33.54 | 33.58 |
| final model size (MB) | 16.3 | 16.4 | 16.6 | 17.1 | 18.0 |

# E   Analysis of training and rendering time

We tested the training and rendering speed of our method and other methods on NHR dataset, where the rendering speed are tested at resolution of 512*384. The full comparison results are shown in Table 3.

# F   Broader impact

Our method can reduce the storage of volumetric video, thereby promoting its practical application. We have not yet seen potential negative impacts. Notably, our method reconstructs volumetric video based on observed multi-view videos, thus it will not be used to generate misleading or false contents.

# G   Licenses

We use NHR and DyNeRF datasets in our experiments. NHR dataset is with License: CC-BY-NC-SA 4.0 and DyNeRF dataset is with License: CC-BY-NC 4.0.

Table 3: **Analysis of training and rendering time on NHR dataset.**

|  | NV | C-NeRF | D-NeRF | DyNeRF | DyMap | K-Planes | Ours |
|---|---|---|---|---|---|---|---|
| Training time (hrs)↓ | >20 | >20 | >20 | >100 | 16 | 2 | 2.5 |
| Rendering time (ms)↓ | 73 | 1969 | 2303 | 5195 | 33 | 384 | 61 |

Table 4: **Quantitative results on NHR dataset.** Note that D-NeRF and DyNeRF are both single MLP based methods, which is of low model size but extremely slow training speed.

| Scene ID | Metrics | NV | C-NeRF | D-NeRF | DyNeRF | DyMap | K-Planes | Ours |
|---|---|---|---|---|---|---|---|---|
| Sport1 | PSNR↑ | 31.76 | 31.81 | 30.12 | 31.76 | 32.92 | 32.23 | 33.51 |
|  | SSIM↑ | 0.951 | 0.954 | 0.934 | 0.954 | 0.959 | 0.958 | 0.957 |
| Sport2 | PSNR↑ | 31.48 | 32.12 | 30.18 | 32.43 | 33.19 | 32.17 | 33.05 |
|  | SSIM↑ | 0.933 | 0.95 | 0.917 | 0.945 | 0.954 | 0.949 | 0.944 |
| Sport3 | PSNR↑ | 31.04 | 31.99 | 29.66 | 31.33 | 33.59 | 30.94 | 32.48 |
|  | SSIM↑ | 0.94 | 0.95 | 0.914 | 0.944 | 0.956 | 0.942 | 0.946 |
| Basketball | PSNR↑ | 29.17 | 29.35 | 27.02 | 27.97 | 29.11 | 29.01 | 29.34 |
|  | SSIM↑ | 0.938 | 0.942 | 0.914 | 0.929 | 0.943 | 0.938 | 0.939 |
|  | Size (MB)↓ | 658 | 1019 | 4 | 12 | 239 | 103 | 16.6 |

Table 5: **Quantitative results on DyNeRF dataset.** DyNeRF[1] and LLFF[1] only report metrics on the Flame Salmon scene.

| Scene ID | Metrics | LLFF[1] | DyNeRF[1] | Mixvoxels | K-Planes | Ours |
|---|---|---|---|---|---|---|
| Coffee Martini | PSNR↑ | - | - | 29.36 | 29.1225 | 28.86 |
|  | SSIM↑ | - | - | 0.946 | 0.905 | 0.898 |
| Cook Spinach | PSNR↑ | - | - | 31.61 | 31.331 | 32.00 |
|  | SSIM↑ | - | - | 0.965 | 0.932 | 0.933 |
| Cut Beef | PSNR↑ |  |  | 31.30 | 31.12 | 31.39 |
|  | SSIM↑ | - | - | 0.965 | 0.934 | 0.928 |
| Flame Salmon | PSNR↑ | 23.24 | 29.58 | 29.92 | 29.14 | 28.94 |
|  | SSIM↑ | 0.848 | 0.961 | 0.945 | 0.905 | 0.893 |
| Flame Steak | PSNR↑ | - | - | 31.21 | 32.03 | 30.34 |
|  | SSIM↑ | - | - | 0.970 | 0.944 | 0.942 |
| Sear Steak | PSNR↑ | - | - | 31.43 | 30.54 | 31.98 |
|  | SSIM↑ | - | - | 0.97 | 0.95 | 0.94 |
|  | Size (MB)↓ | - | 28 | 500 | 103 | 27 |