# OpenReview forum: "Compact Neural Volumetric Video Representations with Dynamic Codebooks"
_NeurIPS.cc/2023/Conference — NeurIPS 2023 poster_

### Official Review · Reviewer_u1Fc · 2023-07-04

**Soundness:** 3 good
**Presentation:** 3 good
**Contribution:** 3 good
**Rating:** 7
**Confidence:** 4

**Summary:**

A method for compressing Volumetric Videos is presented in this work. It is based on NeRF with a factored multi feature plane representation. The features of the model are compressed in two stages. In the first, a codebook for features is constructed based on the average contribution a feature has to the total reconstruction. Features are weighted based on their score contribution to the NeRF integral. Ones with low score contributions are merged into a zero code, the top 30% of codes are retained, and the rest are clustered using an exponential-moving average algorithm. In the second compression stage,  a small number of features per temporal segment are selected based on the accumulated gradients of back propagation on the codebook compressed model. These features are then optimized and added to the codebook, dynamically growing the codebook with time. The resulting method produces reconstruction quality and compression ratios comparable to SOTA while requiring only hours instead of days to train.

**Strengths:**

A simple method that achieves SOTA accuracy and compression performance but is much more efficient.

**Weaknesses:**

- Writing could be a bit clearer
- Only evaluated on two scenes

**Questions:**

- Adding some implementation details where the method is described would help clarify thing sooner. For example, how accumulated score contributions are used to select the three categories, and how a temporal fragment is defined.
- The clustering algorithm in Sec B or supplementary could be clearer and added to the main text. ie. is this performed while training the features as in VQ-VAE, or is performed as a post-processing?


**Limitations:**

Authors do not address limitations.

---

> ### Author Rebuttal · Authors · 2023-08-10
>
> We thank the reviewer for the insightful suggestions. We address the major concerns below:
>
> **Q1: Writing could be a bit clearer.**
>
> **A1:** Thanks for your suggestion. We will improve our writing and include more detailed descriptions of our methods in the revised paper.
>
> **Q2: Only evaluated on two scenes.**
>
> **A2:**: We conducted experiments on two datasets, totaling 10 scenes (4 from the NHR dataset and 6 from the DyNeRF dataset). We have provided the results for each scene in the supplementary materials.
>
> **Q3: Adding some implementation details where the method is described would help clarify thing sooner.**
>
> **A3:** We appreciate your suggestion. In the revised paper, we will include some of the crucial details in the method section, making the paper easier to follow.
>
> **Q4: The clustering algorithm in Sec B or supplementary could be clearer and added to the main text. ie. is this performed while training the features as in VQ-VAE, or is performed as a post-processing?**
>
> **A4:** The clustering algorithm is performed as a post-processing. We will make this part clearer and add them to the main text in the revised paper.

---

### Official Review · Reviewer_Jvh8 · 2023-07-05

**Soundness:** 3 good
**Presentation:** 3 good
**Contribution:** 3 good
**Rating:** 6
**Confidence:** 4

**Summary:**

The paper addresses the challenge of representing high-fidelity volumetric videos with low storage cost. The authors proposed a novel neural representation called the dynamic codebook, which aims to reduce the spatial and temporal redundancy of feature grids inherent to scenes due to self-similarity. It achieves this by merging similar features for model compression and compensating for potential declines in rendering quality through a set of dynamic codes. The experimental results demonstrate that the dynamic codebook method achieves state-of-the-art rendering quality while achieving higher storage efficiency.

The contributions of the paper are twofold:
1. This paper proposes the dynamic codebook representation, which effectively reduces model size by minimizing feature redundancy in space and time while maintaining rendering quality in dynamic detailed regions.
2. Experimental results show that the proposed approach achieves rendering quality comparable to state-of-the-art methods while significantly improving storage efficiency.


**Strengths:**

The paper introduces a novel approach that use the dynamic codebook to reduce the spatial and temporal redundancy.

The paper is well-structured, with clear section headings, and visual aids such as figures are effectively utilized to illustrate key points or enhance comprehension.


**Weaknesses:**

Experimental details and results seem incompletely reported, so I have some confusion about this work. I defer some of my issues in "Questions".

**Questions:**

1. This paper claims that “…which is of low model size but extremely slow training speed.” In Table 1, but there is no mention of the training time for the models. It would be helpful to provide insights into the time required for training the model and how it compares to other models.
2. Regarding the comparison of model size, I noticed that this paper uses “post-processing” including quantization and entropy coding techniques. According to my understanding, this techniques can affect the model size, whether other comparison models use this techniques? If not, it would be beneficial to discuss the potential outcomes if these post-processing steps were not applied.
3. In line 237, this paper uses different k for different dataset. Is there any particular consideration followed while setting the value of k?
In reference to the dynamic codebook's size, it would be useful to explore what constitutes an appropriate size. The codebook-based compression method mentioned can significantly reduce model storage but may result in a decline in rendering quality. Thus, it would be meaningful to investigate how variations in k affect the same dataset.
4. The model size is composed of the quantized dynamic codebook, index planes as well as MLP parameters with entropy encoding, so the bit allocation of different components of the model should be provided, which can help the understanding of the proposed methodology.



**Limitations:**

The authors have partially addressed the limitations of existing work, though there is space for improvement (see the section Weaknesses and Questions).

---

> ### Author Rebuttal · Authors · 2023-08-10
>
> We thank the reviewer for the insightful suggestions. We address the major concerns below:
>
> **Q1: It would be helpful to provide insights into the time required for training the model and how it compares to other models.**
>
> **A1:** In Section 4, we mentioned the details of training time: "We train the model on a single NVIDIA A100 GPU, which takes about 1.5 / 4.3 hours for training and 1.0 / 1.7 hours for the construction of a dynamic codebook on NHR / DyNeRF datasets." And we provide comparison to other models here.
>
> |Method             |NV |C-NeRF|D-NeRF|DyNeRF|DyMap|K-Planes|Ours|
> |-------------------|---|------|------|------|-----|--------|----|
> |Training time (hrs)|>20|>20   |>20   |>100  |16   |2       |2.5 |
>
> **Q2: Analysis of post-processing and allocation of model storage.**
>
> **A2:** To more clearly analyse the model storage in our method, we examined the allocation of model storage size at each step on the NHR dataset, in accordance with the sequence of method implementation. Given that our baseline has fewer MLP parameters, we did not further compress the MLP parameters. The primary storage of the initial model (1) lies in the feature planes. We compressed these using a representation of codebook + index planes (2), which significantly reduces storage. We then utilized quantization (3) and entropy encoding (4) techniques for post-processing, achieving further reductions. The results at this stage correspond to the "compression w/o DC" in Table 3 of our paper. Finally, we employed a dynamic codebook (5) to introduce additional storage for codebook and index planes, enhancing the rendering quality. We will further refine these experiments and analysis and incorporate them into the revised paper.
>
> |   |feature planes|codebook|index planes|MLP |Sum   |
> |---|--------------|--------|------------|----|------|
> |1  |119.92        |0       |0           |0.10|120.02|
> |2  |0             |29.04   |0.08        |0.10|29.22 |
> |3  |0             |7.52    |0.08        |0.10|7.70  |
> |4  |0             |5.51    |0.08        |0.10|5.69  |
> |5  |0             |15.50   |0.98        |0.10|16.58 |
>
> **Q3: Analysis of hyperparameters such as k (codebook size).**
>
> **A3:** Our method introduces some new hyperparameters such as k. However, there are some principles to follow. Here, we outline some guidelines for setting these hyperparameters:
>
> - **Size of the codebook ('k'):** The setting of the codebook size relates to the trade-off between storage size and rendering quality. For more content-rich scenarios like DyNeRF, a larger codebook is needed. On the other hand, for relatively simpler scenarios like NHR, a smaller codebook suffices.
> - **The ratio of codes to discard or retain during codebook compression:** We retain the top 30% of codes with the highest importance score contributions in all datasets. These contribute approximately 80% of the total importance score. Setting the threshold based on this ratio allows us to substantially reduce storage while trying to maintain the original rendering quality.
>
> We designed corresponding ablation studies on NHR, and the results are as follows:
>
> |percent of retained code   |10   |20   |30   |
> |---------------------------|-----|-----|-----|
> |PSNR (w/o dynamic codebook)|32.39|32.57|32.85|
> |PSNR (w/ dynamic codebook) |33.40|33.46|33.51|
> |final model size (MB)      |16.3 |16.5 |16.6 |
>
> |k                          |1024 |2048 |4096 |8192 |16384|
> |---------------------------|-----|-----|-----|-----|-----|
> |PSNR (w/o dynamic codebook)|32.09|32.47|32.85|32.89|33.01|
> |PSNR (w/ dynamic codebook) |33.30|33.43|33.51|33.54|33.58|
> |final model size (MB)      |16.3 |16.4 |16.6 |17.1 |18.0 |
>
> The results evidents that both codebook size and the retention ratio influence the storage size and rendering quality of our method. However, the impact is not substantial, indicating that our method is fairly robust to these hyperparameters.
>
> We will design and conduct more comprehensive ablation studies, which will be incorporated into the revised paper.

---

> > ### Comment · Reviewer_Jvh8 · 2023-08-19
> >
> > Thank you for the rebuttal. All my concerns were well addressed.

---

> > > ### Author Response · Authors · 2023-08-19
> > >
> > > Thank the reviewer for acknowledging our rebuttal. Your feedback is very constructive, and we will revise the paper according to it.

---

### Official Review · Reviewer_ZNi1 · 2023-07-05

**Soundness:** 3 good
**Presentation:** 3 good
**Contribution:** 2 fair
**Rating:** 4
**Confidence:** 4

**Summary:**

The proposed method applies the codebook technique to explicit feature plane representation and a dynamic codebook is further proposed for dynamic scenes. Experimental results demonstrate the good performance of the proposed methods.

**Strengths:**

1. The proposed method significantly reduces the model size while keeping a similar render quality with K-Planes [35] from both qualitative metrics and visualization.
2. The proposed dynamic codebook is easy to follow and the problem addressed is important and interesting.


**Weaknesses:**

1. Considering that previous work VQRF has introduced codebook to feature grid representation for reducing the model size with an excellent compression ratio, the proposed method here that introduces codebook to feature plane representation is reasonable but not of sufficient novelty.
2. The main contribution of model size reduction comes from the codebook technique. The efficiency of the proposed dynamic codebook is somewhat limited.
3. There is no experimental comparison of the time cost. The influence of the optimization process caused by the dynamic codebook is unclear.
4. There lack some explanation and evaluation about some hyperparameters and settings. For example, In Sec 4, Line 235: "We then retain the top 30% of codes with the highest importance score contributions and cluster the remaining codes into k codes". The reason for the setting 30% and the influence of k can be provided for better demonstrating the efficiency of the proposed strategy.


**Questions:**

1. The proposed dynamic codebook should be applicable to other explicit representations, e.g., feature grids. The authors may explain the reason why they did not conduct related experiments.

---

> ### Author Rebuttal · Authors · 2023-08-10
>
> We thank the reviewer for the insightful suggestions. We address the major concerns below:
>
> **Q1: Introducing codebook to feature plane representation is reasonable but not of sufficient novelty.**
>
> **A1:** We would like to emphasize that we have two core contributions, which make our method distinct from previous works:
>
> - **Technical contributions.** We present a carefully-designed method for volumetric video compression. We claim that directly applying compression methods for static scenes to dynamic scenes will result in considerable information loss. This is because they do not take into account the temporal variability characteristics of dynamic scenes, which is fundamentally different from static scenes. To this end, we designed a dynamic codebook compression method tailored to the characteristics of dynamic scenes. Our approach identifies areas that most require enhancement in each time fragment and incrementally supplements codes into dynamic codebook.
> - **Experimental contributions.** Empirically, we discovered that simply applying the codebook compression methods from static scenes to dynamic scenes results in a noticeable decline in quality in detailed regions, such as facial areas. To overcome this problem, we implemented the proposed dynamic codebook with thoughtful method design and engineering efforts, which achieved a high compression rate on two representative and challenging dynamic scene datasets (NHR and DyNeRF) while ensuring rendering quality comparable to state-of-the-art (SOTA) methods.
>
> We believe that the contributions mentioned above will bring new insights to this field and benefit the community.
>
> **Q2: The effectiveness of the proposed dynamic codebook is somewhat limited.**
>
> **A2:** The dynamic codebook can compensate for the quality loss resulting from compression while requiring less storage. We have provided corresponding qualitative and quantitative analyses in Figure 4 and Table 3. The qualitative results clearly show that compression leads to significant quality loss in detailed areas such as the face, while the dynamic codebook can improve the quality in these areas. From a quantitative perspective, the PSNR improvement brought by the dynamic codebook is not particularly large. We claim there are two reasons for this: 1. The purpose of our method is to compress while maintaining rendering quality, so the rendering quality of the baseline before compression can be considered an upper limit that we can hardly exceed. 2. Detailed areas occupy a smaller proportion of the image, while PSNR is averaged over all pixels when calculated, so the improvement in the quality of detailed areas is not very pronounced in terms of PSNR enhancement.
>
> **Q3: The influence of the optimization process caused by the dynamic codebook is unclear.**
>
> **A3:** In Section 4, we mentioned the details of training time: "We train the model on a single NVIDIA A100 GPU, which takes about 1.5 / 4.3 hours for training and 1.0 / 1.7 hours for the construction of a dynamic codebook on NHR / DyNeRF datasets."
>
> **Q4: There lack some explanation and evaluation about some hyperparameters and settings.**
>
> **A4:** Our method indeed introduces some new hyperparameters. However, there are some principles to follow. Here, we outline some guidelines for setting these hyperparameters:
>
> - **Size of the codebook ('k'):** The setting of the codebook size relates to the trade-off between storage size and rendering quality. For more content-rich scenarios like DyNeRF, a larger codebook is needed. On the other hand, for relatively simpler scenarios like NHR, a smaller codebook suffices.
> - **The ratio of codes to discard or retain during codebook compression:** We retain the top 30% of codes with the highest importance score contributions in all datasets. These contribute approximately 80% of the total importance score. Setting the threshold based on this ratio allows us to substantially reduce storage while trying to maintain the original rendering quality.
>
> We designed corresponding ablation studies on NHR, and the results are as follows:
>
> |percent of retained code   |10   |20   |30   |
> |---------------------------|-----|-----|-----|
> |PSNR (w/o dynamic codebook)|32.39|32.57|32.85|
> |PSNR (w/ dynamic codebook) |33.40|33.46|33.51|
> |final model size (MB)      |16.3 |16.5 |16.6 |
>
> |k                          |1024 |2048 |4096 |8192 |16384|
> |---------------------------|-----|-----|-----|-----|-----|
> |PSNR (w/o dynamic codebook)|32.09|32.47|32.85|32.89|33.01|
> |PSNR (w/ dynamic codebook) |33.30|33.43|33.51|33.54|33.58|
> |final model size (MB)      |16.3 |16.4 |16.6 |17.1 |18.0 |
>
> The results evidents that both codebook size and the retention ratio influence the storage size and rendering quality of our method. However, the impact is not substantial, indicating that our method is fairly robust to these hyperparameters.
>
> We will design and conduct more comprehensive ablation studies, which will be incorporated into the revised paper.
>
> **Q5: Why not conduct experiments to other explicit representations, e.g., feature grids?**
>
> **A5:** 4D feature grid requires too large memory, which is infeasible, and there is currently no such work. At present, there are only two types of dynamic scene reconstruction/view synthesis work based on feature grids: 1. those based on feature planes (2D grid) 2. those using a canonical 3D feature grid in combination with deformation field.
>
> Our work focuses on compression based on the first type of method. The second type of method, is actually a technique for compressing temporal information with deformation. However, it is not capable of dealing with some complex dynamic scenes (some complex changes or movements cannot be well modeled using deformation).

---

> > ### Comment · Reviewer_ZNi1 · 2023-08-18
> >
> > Thank you for your response. Your responses clarified my main concerns about these 2 hyperparameters of the dynamic codebook. However, I am still concerned about this approach that achieves better render quality via sacrificing training time, there is only a marginal quality improvement. The article claimed to solve issues on detailed regions such as facial areas but lacked details. To better discuss this topic, more specifics on the causes, more visual examples, and quantitative analysis showing improvements are needed.
> >
> > After a re-evaluation of your paper, I willl keep my rating.

---

> > > ### Author Response · Authors · 2023-08-21
> > >
> > > Thanks for your feedback. To better demonstrate our improvements in detailed areas such as the facial areas, we conduct an additional ablation study. We segment the facial area of each image on the NHR dataset using SCHP (Self Correction for Human Parsing) and evaluate the PSNR for these areas. The results are as follows:
> > >
> > > | |Sport1|Sport2|Sport3|Basketball|Average|
> > > |-|-|-|-|-|-|
> > > |baseline          |21.02|22.84|21.59|23.25|22.18|
> > > |compression w/o DC|20.08|21.51|20.89|21.53|21.00|
> > > |compression w/ DC |20.84|22.54|21.50|22.75|21.91|
> > >
> > > The results indicate that direct compression leads to significant degradation in rendering quality, while the dynamic codebook can substantially compensate for this loss. We will conduct more comprehensive experiments and include more qualitative results in the revised paper.

---

### Official Review · Reviewer_bWLa · 2023-07-06

**Soundness:** 4 excellent
**Presentation:** 3 good
**Contribution:** 4 excellent
**Rating:** 7
**Confidence:** 4

**Summary:**

The authors of this paper propose  a dynamic codebook, which optimizes away codes of low importance to rendering the scene, clusters 70% of the least importance codes and optimizes the remaining codes for every time fragment. The process the authors introduce are intuitive and each step furthers compression or works towards reducing the distortion introduced in the compression process.

I have read and acknowledged the author's rebuttal and will remain with my generally positive assessment.

**Strengths:**

The introduction of a dynamic codebook is intuitive and handles a number of problems in the codebook creation process. First keeping highly important codes, clustering low important codes to a zero token, clustering remaining codes and then using gradient aware code optimization to further increase details for specific time fragments.

The authors demonstrate near state of the art PSNR on NHR and DyNeRF datasets ( < 0.25 dB PSNR difference) with substantially (more than 14x) smaller models.

**Weaknesses:**

As discussed by the authors, the dynamic codebook construction takes at least an hour for NHR and almost two hours for DyNeRF and substantially increases the total end to end time and rendering time.

The dynamic codebook introduces more hyperparameters to tune, what percentage of features to keep, drop, cluster and how many dynamic codewords get added. NHR and DyNeRF seem to differ substantially in these and it's unclear how much of this is based off of framerate, resolution, or complexity.

**Questions:**

1. In L169, the authors state that 99.9% of the importance is contributed by merely 10% of the voxels in the 2D case and that it is not as bad in the 3D case. Is there a citation or an empirical result the authors could share on this?

2. How large are the codebooks for each of the time fragments? In L237, the authors state that the clustered codes are set to 4096 (NHR) and 16,384 for DyNeRF, but not how many codes are being optimized by the dynamic codebook for each fragment.

3. How much experimentation did it take to discover the different hyperparameters for the dynamic codebook? Is it easy to extrapolate to other datasets? Do much more could be gained by even more optimal hyperparameters on a nerf by nerf basis?

4. Any qualitative or user study on different nerfs? The quality difference seems minor, so it's not clear if it's visually preferred over other methods.

Typo in Table 3:
basline => baseline

**Limitations:**

No concerns

---

> ### Author Rebuttal · Authors · 2023-08-10
>
> We thank the reviewer for the insightful suggestions. We address the major concerns below:
>
> **Q1: The dynamic codebook construction increases the total end to end time and rendering time.**
>
> **A1:** Firstly, the dynamic codebook has a slight influence on the rendering time as the indexing process of the codebook is nearly instantaneous. Secondly, although the training time increases by 1-2 hours, the overall training time remains comparable to other methods based on feature grids, and is at least an order of magnitude faster than methods like DyNeRF. Moreover, the benefits are significant, especially in terms of storage efficiency improvement. This presents substantial application value in scenarios where high storage efficiency is required but sensitivity to training time is not a concern.
>
> We also provide comparison results of training and rendering time with SOTA methods, where rendering speed is tested on NHR dataset at resolution of 512*384:
>
> |Method             |NV |C-NeRF|D-NeRF|DyNeRF|DyMap|K-Planes|Ours|
> |-------------------|---|------|------|------|-----|--------|----|
> |Training time (hrs)|>20|>20   |>20   |>100  |16   |2       |2.5 |
> |Rendering time (ms)|73 |1969  |2303  |5195  |33   |384     |61  |
>
> **Q2: The dynamic codebook introduces more hyperparameters, and it's unclear how to adjust them.**
>
> **A2:** Our method indeed introduces some new hyperparameters. However, there are some principles to follow. Here, we outline some guidelines for setting these hyperparameters:
>
> - **Size of the codebook ('k'):** The setting of the codebook size relates to the trade-off between storage size and rendering quality. For more content-rich scenarios like DyNeRF, a larger codebook is needed. On the other hand, for relatively simpler scenarios like NHR, a smaller codebook suffices.
> - **The ratio of codes to discard or retain during codebook compression:** We retain the top 30% of codes with the highest importance score contributions in all datasets. These contribute approximately 80% of the total importance score. Setting the threshold based on this ratio allows us to substantially reduce storage while trying to maintain the original rendering quality.
>
> We designed corresponding ablation studies on NHR, and the results are as follows:
>
> |percent of retained code   |10   |20   |30   |
> |---------------------------|-----|-----|-----|
> |PSNR (w/o dynamic codebook)|32.39|32.57|32.85|
> |PSNR (w/ dynamic codebook) |33.40|33.46|33.51|
> |final model size (MB)      |16.3 |16.5 |16.6 |
>
> |k                          |1024 |2048 |4096 |8192 |16384|
> |---------------------------|-----|-----|-----|-----|-----|
> |PSNR (w/o dynamic codebook)|32.09|32.47|32.85|32.89|33.01|
> |PSNR (w/ dynamic codebook) |33.30|33.43|33.51|33.54|33.58|
> |final model size (MB)      |16.3 |16.4 |16.6 |17.1 |18.0 |
>
> The results evidents that both codebook size and the retention ratio influence the storage size and rendering quality of our method. However, the impact is not substantial, indicating that our method is fairly robust to these hyperparameters.
>
> We will design and conduct more comprehensive ablation studies, which will be incorporated into the revised paper.
>
> **Q3: The basis of the distribution of importance score.**
>
> **A3:** The statement "when representing 3D scenes using feature grids, 99.9% of the importance is contributed by merely 10% of the voxels" originates from reference [16], which focused on compressing static scenes. In our paper, we claim that "the situation is not as extreme when we use 2D planes to represent volumetric videos" primarily for two reasons: 1. Volumetric videos have much richer content compared to static scenes. 2. The representation using feature planes is somewhat more compact than using a 3D feature grid. Hence, when we use feature planes to represent volumetric videos, the situation isn't as extreme as when using feature grids for static scenes. Moreover, our empirical findings support this claim--we discovered that 99.9% of the importance is contributed by approximately 75% of the voxels and 80% of the importance is contributed by approximately 30% of the voxels.
>
> **Q4: How many codes are being optimized by the dynamic codebook for each fragment?**
>
> **A4:** In section 4, we mentioned that "we optimize 1000 appearance codes and 5000 density codes for each time fragment."
>
> **Q5: Any qualitative or user study on different nerfs? The quality difference seems minor, so it's not clear if it's visually preferred over other methods.**
>
> **A5:** We provide qualitative comparisons with other methods in Figure 2 and Figure 3. The quality difference between our approach and the state-of-the-art methods is indeed minor. However, this is not conflict with our claimed contribution: we can achieve rendering quality comparable to state-of-the-art methods, but with a significant boost in storage efficiency. The improvement of storage efficiency can be clearly seen from the quantitative results in Table 1 and Table 2.

---

> > ### Comment · Reviewer_bWLa · 2023-08-18
> > **Response**
> >
> > Thank you for the detailed responses to my questions and concerns as this provides better clarity of your work. I'm still happy with my recommendation of Accept  and believe this is a sufficiently strong technical work for NeurIPS.

---

> > > ### Author Response · Authors · 2023-08-19
> > >
> > > Thanks for your response. We will revise the paper according to your suggestions.

---

### Official Review · Reviewer_uPwZ · 2023-07-09

**Soundness:** 3 good
**Presentation:** 3 good
**Contribution:** 2 fair
**Rating:** 3
**Confidence:** 5

**Summary:**

This paper presents a novel approach for representing volumetric video using a dynamic codebook that incorporates the temporal correlation of features. This addresses the drawback of existing feature grid-based methods, which overlook this correlation.

**Strengths:**

1. The proposed method uses a multidimensional feature space and a dynamic codebook to model the changing scene.
2. The paper presents model compression techniques, including pruning and weight clustering, which reduce the size of the model.

**Weaknesses:**

1. The article's novelty is limited as the introduction of the dynamic codebook approach appears to be primarily focused on engineering aspects. Additionally, the results from ablation experiments suggest that this approach is not particularly effective, and although a large number of model parameters are introduced, there is only a marginal improvement in video quality.

2. It is impractical to manually adjust dynamic codebook parameters (such as k) for different videos in real-world applications.

3. The paper lacks comparison results regarding training and rendering time.

**Questions:**

1. What specific improvements have been made in Sections 3.1 and 3.2 compared to existing works, like [16]?
2. Why were certain methods not tested on the DyNeRF dataset?

**Limitations:**

Further research is needed to develop effective methodologies for dynamic codebooks. Although the paper enhances the expressive capabilities of the dynamic neural representation model, there is room for future investigation into optimizing training and rendering speed.

---

> ### Author Rebuttal · Authors · 2023-08-10
>
> We thank the reviewer for the insightful suggestions. We address the major concerns below:
>
> **Q1: Limited novelty and no apparent improvements compared to existing works.**
>
> **A1:** We would like to emphasize that we have two core contributions, which make our method distinct from previous works:
> - **Technical contributions.** We present a carefully-designed method for volumetric video compression. We build our method on feature plane instead of 3D feature grid, which is a more suitable representation for volumetric videos. We claim that directly applying compression methods for static scenes to dynamic scenes will result in considerable information loss. This is because they do not take into account the temporal variability characteristics of dynamic scenes, which is fundamentally different from static scenes. To this end, we designed a dynamic codebook compression method tailored to the characteristics of dynamic scenes. Our approach identifies areas that most require enhancement in each time fragment and incrementally supplements codes into dynamic codebook.
> - **Experimental contributions.** Empirically, we discovered that simply applying the codebook compression methods from static scenes to dynamic scenes results in a noticeable decline in quality in detailed regions, such as facial areas. To overcome this problem, we implemented the proposed dynamic codebook with thoughtful method design and engineering efforts, which achieved a high compression rate on two representative and challenging dynamic scene datasets (NHR and DyNeRF) while ensuring rendering quality comparable to state-of-the-art (SOTA) methods.
>
> We believe that the contributions mentioned above will bring new insights to this field and benefit the community.
>
> **Q2: The results from ablation experiments suggest that this approach is not particularly effective.**
>
> **A2:** The dynamic codebook can compensate for the quality loss resulting from compression while requiring less storage. We have provided corresponding qualitative and quantitative analyses in Figure 4 and Table 3. The qualitative results clearly show that compression leads to significant quality loss in detailed areas such as the facial regions, while the dynamic codebook can improve the quality in these areas. From a quantitative perspective, the PSNR improvement brought by the dynamic codebook is not particularly large. We claim there are two reasons for this: 1. The purpose of our method is to compress while maintaining rendering quality, so the rendering quality of the baseline before compression can be considered an upper limit that we can hardly exceed. 2. Detailed areas occupy a small proportion of the image, while PSNR is averaged over all pixels, so the improvement in detailed areas is not very pronounced in terms of PSNR enhancement.
>
> **Q3: It is impractical to manually adjust the hyperparameters in real-world applications.**
>
> **A3:** Our method indeed introduces some new hyperparameters. However, there are some principles to follow. Here, we outline some guidelines for setting these hyperparameters:
>
> - **Size of the codebook ('k'):** The setting of the codebook size relates to the trade-off between storage size and rendering quality. For more content-rich scenarios like DyNeRF, a larger codebook is needed. On the other hand, for relatively simpler scenarios like NHR, a smaller codebook suffices.
> - **The ratio of codes to discard or retain during codebook compression:** We retain the top 30% of codes with the highest importance score contributions in all datasets. These contribute approximately 80% of the total importance score. Setting the threshold based on this ratio allows us to substantially reduce storage while trying to maintain the original rendering quality.
>
> We designed corresponding ablation studies on NHR, and the results are as follows:
>
> |percent of retained code   |10   |20   |30   |
> |---------------------------|-----|-----|-----|
> |PSNR (w/o dynamic codebook)|32.39|32.57|32.85|
> |PSNR (w/ dynamic codebook) |33.40|33.46|33.51|
> |final model size (MB)      |16.3 |16.5 |16.6 |
>
> |k                          |1024 |2048 |4096 |8192 |16384|
> |---------------------------|-----|-----|-----|-----|-----|
> |PSNR (w/o dynamic codebook)|32.09|32.47|32.85|32.89|33.01|
> |PSNR (w/ dynamic codebook) |33.30|33.43|33.51|33.54|33.58|
> |final model size (MB)      |16.3 |16.4 |16.6 |17.1 |18.0 |
>
> The results evidents that both codebook size and the retention ratio influence the storage size and rendering quality of our method. However, the impact is not substantial, indicating that our method is fairly robust to these hyperparameters.
>
> We will design and conduct more comprehensive ablation studies, which will be incorporated into the revised paper.
>
> **Q4: The paper lacks comparison results regarding training and rendering time.**
>
> **A4:** In Section 4, we mentioned the details of training time: "We train the model on a single NVIDIA A100 GPU, which takes about 1.5 / 4.3 hours for training and 1.0 / 1.7 hours for the construction of a dynamic codebook on NHR / DyNeRF datasets." The rendering time depends on factors such as the complexity of the scene and the resolution of the image. We tested the rendering speed of our method and other methods on NHR dataset at resolution of 512*384. The full comparison results are as follows:
>
> |Method             |NV |C-NeRF|D-NeRF|DyNeRF|DyMap|K-Planes|Ours|
> |-------------------|---|------|------|------|-----|--------|----|
> |Training time (hrs)|>20|>20   |>20   |>100  |16   |2       |2.5 |
> |Rendering time (ms)|73 |1969  |2303  |5195  |33   |384     |61  |
>
> **Q5: Why were certain methods not tested on the DyNeRF dataset?**
>
> **A5:** Based on the results reported in DyNeRF, the performance of Neural Volumes is quite poor. C-NeRF, as a follow-up to Neural Volumes, is also expected to perform poorly, and they both require substantial storage. As for DyMAP, we ran it ourselves and found that it does not work on the DyNeRF dataset.

---

> > ### Comment · Reviewer_uPwZ · 2023-08-20
> > **response**
> >
> > Thank you for the author's response. The author has provided an implicit compression method with a dynamic codebook. However, from the implementation results, the method may be not convincing:
> > The baseline model used for comparison (such as DyNeRF) is uncompressed, and previous compression works (such as VQRF) have already provided efficient codebook representations and high compression ratios.
> > Therefore, even without introducing a dynamic codebook, it is possible to achieve better performance than the baseline model. This can be inferred from the results of Ablation studies (Table 3 compression w/o DC).
> > Furthermore, Ablation studies are confusing. Using DC slightly increases PSNR but significantly increases model size. This result does not demonstrate the superiority of the compression technique, even though it still reduces model size compared to the baseline.
> > Moreover, in terms of technical details, manually setting hyperparameters (k) is impractical and unfeasible in real-world applications. The author's response in rebuttal did not provide a clear explanation for this issue.

---

> > > ### Author Response · Authors · 2023-08-21
> > >
> > > Thank the reviewer for the response. We would like to clarify the validity and practicality of our method as following:
> > >
> > > 1. **Comparison with baseline models**.
> > >     - **DyNeRF**. The model of DyNeRF is an MLP, which is challenging to further compress. The primary issue of DyNeRF is its slow training and rendering speed.
> > >     - **K-Planes**. We have compressed the important baseline, K-Planes, for comparison, which is referred to as `compression w/o DC` in our ablation studies.
> > > 2. **Performance of VQRF**. Using VQRF for dynamic scene compression results in quality loss in detailed areas, with a PSNR decrease of about 1 point (please refer to the table below for more details).
> > > 3. **Effectiveness of DC**. DC can improve the PSNR by approximately 0.91 in detailed areas. As mentioned in the paper, DC primarily enhances the rendering quality in detailed areas such as facial areas. Since these areas have a small pixel count proportionally, they don't significantly impact the overall PSNR. However, they greatly affect the rendering quality, as shown in Figure 4 of the paper. To better illustrate this, we conducted a quantitative ablation study on the facial areas. Specifically, we segment the facial area of each image on the NHR dataset using SCHP (Self Correction for Human Parsing) and evaluate the PSNR for these areas. The results are as follows:
> > >
> > > | |Sport1|Sport2|Sport3|Basketball|Average|
> > > |-|-|-|-|-|-|
> > > |baseline          |21.02|22.84|21.59|23.25|22.18|
> > > |compression w/o DC|20.08|21.51|20.89|21.53|21.00|
> > > |compression w/ DC |20.84|22.54|21.50|22.75|21.91|
> > >
> > > 4. **Hyperparameters**. Through our ablation study on hyperparameters, we demonstrated that the impact of hyperparameters on our method is low. Furthermore, as mentioned in our rebuttal, there are some clear principles for setting the hyperparameters. Therefore, we believe our method is practical for real-world applications. We will further revise the principles and the ablation analysis and incorporate them into paper.

---

> > > > ### Comment · Reviewer_uPwZ · 2023-08-21
> > > > **response**
> > > >
> > > > Thank you for your response. There are a few concerns regarding the paper's method that need to be addressed. Firstly, if the proposed method only enhances performance in specific facial regions, it may not be sufficient to demonstrate its overall effectiveness. Secondly, it would be helpful to provide a specific numerical value for the size of the model used in the ablation study when discussing compression with DC method. Without this information, it is difficult to fully evaluate the effectiveness of your approach. Additionally, I noticed that Table 2 of your experiment data does not show any improvement in SSIM values compared to other baseline methods; instead, there appears to be a significant lag behind them. This discrepancy is quite perplexing.

---

> > > > > ### Author Response · Authors · 2023-08-21
> > > > >
> > > > > Thanks for your response. We would like to claim that:
> > > > > 1. Our method is capable of enhancing the rendering quality in all detailed areas, not just the facial areas. We conducted quantitative experiments using the facial areas as an example to demonstrate this. This does not mean that we did not achieve improvements in other detailed areas. We will provide more comprehensive experiments and analysis in the revised paper.
> > > > > 2. We have already provided detailed quantitative data on model sizes in Table 1 of our paper. We excerpt the relevant data as follows:
> > > > >
> > > > > ||NHR|DyNeRF|
> > > > > |-|-|-|
> > > > > |baseline|91.0 MB|523 MB|
> > > > > |compression w/o DC|5.7 MB|22 MB|
> > > > > |compression w/ DC|16.6 MB|27 MB|
> > > > > 3. We chose K-Planes as our baseline. Our final SSIM is almost identical to the baseline, and there is a significant improvement in storage efficiency, which strongly supports our claim. The SSIM of Mixvoxels is 0.3-0.4 higher than both K-Planes and ours, but its storage size is much larger (500MB). We consider this to be reasonable.

---

> > > > > > ### Comment · Reviewer_uPwZ · 2023-08-21
> > > > > > **response**
> > > > > >
> > > > > > Thank you for your response. The SSIM values of K-Planes and your approach are 0.927 and 0.923, respectively. I would like to point out that these values are not identical, indicating a significant gap that cannot be ignored. It may be worth investigating the low-level vision task, such as image super-resolution, where the latest state-of-the-art (SOTA) methods typically show an improvement of 0.001 in SSIM compared to previous SOTA approaches.
> > > > > >
> > > > > > Furthermore, in your response, it is evident that the model size for compression with dynamic code (DC) is 16.6MB, whereas the corresponding model size for compression without DC is only 5.7MB. This suggests that you have used a larger model to achieve better performance; therefore, it cannot be concluded that your approach is effective solely based on this comparison alone. Generally speaking, using more parameters often leads to improved results even with the same architecture. To clarify further: if your approach utilizes either the same or fewer parameters than compression without DC while achieving superior performance, then it would provide stronger evidence supporting your idea.
> > > > > >
> > > > > > Another aspect I find perplexing is why compression with DC employs more parameters when you claim that dynamic codebook reduces model size? Additionally, according to the ablation study conducted in this paper itself, it shows that the proposed method without DC actually uses fewer parameters than its counterpart with DC implementation—this inconsistency adds confusion.
> > > > > >
> > > > > > Taking into account all these comments along with other reviewers' opinions, I believe this paper should be rejected due to numerous confusing results.

---

> > > > > > > ### Author Response · Authors · 2023-08-21
> > > > > > >
> > > > > > > Thanks for your response. We would like to claim that:
> > > > > > > 1. Compared to the baseline (K-Planes), our method significantly improves storage efficiency. A decrease of 0.004 in SSIM on this basis should be acceptable. We believe that our method's effectiveness should not be negated based on this decline.
> > > > > > > 2. We did not claim that DC could further reduce storage on top of direct compression. According to our claim in the paper, DC can compensate for the loss of rendering quality in detailed areas caused by direct compression, without introducing significant additional storage. In other words, the storage efficiency compared to K-Planes remains vastly superior after using DC.

---

### Decision · Program_Chairs · 2023-09-21

**Decision:**

Accept (poster)

**Comment:**

The manuscript is focused on minimizing the size of a NeRF model represented by k-planes and has garnered mixed reviews. Three out of five reviewers are supportive, but two have raised concerns about the originality of using codebooks for compression, the necessity for manual hyperparameter tuning, and the absence of a comparison between training and rendering times. Notably, one reviewer found the experimental results ambiguous even after the authors' rebuttal. The AC concurs that the manuscript necessitates significant revisions, especially in terms of training and rendering times (since trading time for space is typically deemed suboptimal, making the inclusion of a time comparison crucial). After consultation with the SAC, the decision is to recommend the manuscript for acceptance. Nevertheless, the AC strongly advises the authors to undertake the necessary revisions for the camera-ready version of the paper.